# Cardiovascular Disease-Associated MicroRNA Dysregulation during the First Trimester of Gestation in Women with Chronic Hypertension and Normotensive Women Subsequently Developing Gestational Hypertension or Preeclampsia with or without Fetal Growth Restriction

**DOI:** 10.3390/biomedicines10020256

**Published:** 2022-01-25

**Authors:** Ilona Hromadnikova, Katerina Kotlabova, Ladislav Krofta

**Affiliations:** 1Department of Molecular Biology and Cell Pathology, Third Faculty of Medicine, Charles University, 100 00 Prague, Czech Republic; katerina.kotlabova@lf3.cuni.cz; 2Third Faculty of Medicine, Institute for the Care of the Mother and Child, Charles University, 147 00 Prague, Czech Republic; ladislav.krofta@upmd.eu

**Keywords:** cardiovascular microRNAs, early gestation, expression, gestational hypertension, chronic hypertension, prediction, preeclampsia, screening, whole peripheral venous blood

## Abstract

The aim of the study was to assess if cardiovascular disease-associated microRNAs would be able to predict during the early stages of gestation (within 10 to 13 weeks) subsequent onset of hypertensive pregnancy-related complications: gestational hypertension (GH) or preeclampsia (PE). Secondly, the goal of the study was to assess if cardiovascular disease-associated microRNAs would be able to detect the presence of chronic hypertension in early pregnancies. The retrospective study was performed on whole peripheral blood samples collected from singleton Caucasian pregnancies within the period November 2012 to March 2020. The case control study, nested in a cohort, involved all women with chronic hypertension (*n* = 29), all normotensive women that later developed GH (*n* = 83) or PE with or without fetal growth restriction (FGR) (*n* = 66), and 80 controls selected on the base of equal sample storage time. Whole peripheral blood profiling was performed with the selection of 29 cardiovascular disease-associated microRNAs using real-time RT-PCR. Upregulation of miR-1-3p (51.72% at 10.0% FPR) was observed in patients with chronic hypertension only. Upregulation of miR-20a-5p (44.83% and 33.33% at 10.0% FPR) and miR-146a-5p (65.52% and 42.42% at 10.0% FPR) was observed in patients with chronic hypertension and normotensive women with later occurrence of PE. Upregulation of miR-181a-5p was detected in normotensive women subsequently developing GH (22.89% at 10.0% FPR) or PE (40.91% at 10.0% FPR). In a part of women with subsequent onset of PE, upregulation of miR-143-3p (24.24% at 10.0% FPR), miR-145-5p (21.21% at 10.0% FPR), and miR-574-3p (27.27% at 10.0% FPR) was also present. The combination of microRNA biomarkers (miR-20a-5p, miR-143-3p, miR-145-5p, miR-146a-5p, miR-181a-5p, and miR-574-3p) can predict the later occurrence of PE in 48.48% of pregnancies at 10.0% FPR in early stages of gestation. The combination of upregulated microRNA biomarkers (miR-1-3p, miR-20a-5p, and miR-146a-5p) is able to identify 72.41% of pregnancies with chronic hypertension at 10.0% FPR in early stages of gestation. Cardiovascular disease-associated microRNAs represent promising biomarkers with very good diagnostical potential to be implemented into the current first trimester screening program to predict later occurrence of PE with or without FGR. The comparison of the predictive results of the routine first trimester screening for PE and/or FGR based on the criteria of the Fetal Medicine Foundation and the first trimester screening for PE wo/w FGR using a panel of six cardiovascular disease-associated microRNAs only revealed that the detection rate of PE increased 1.45-fold (48.48% vs. 33.33%).

## 1. Introduction

Clinical risk assessment for preeclampsia (PE) are based on the recommendations of the American College of Obstetricians and Gynecologists (ACOG) and the Society for Maternal-Fetal Medicine [1]. Low dose aspirin (ASA, 75–100 mg) administration from 12 weeks of gestation is recommended if the patient has one or more of the following high risk factors (history of PE, multifetal gestation, chronic hypertension, type 1 or 2 diabetes, renal disease, autoimmune disease such as systemic lupus erythematosus or antiphospholipid syndrome) or if the patient has more than one of these moderate risk factors (nulliparity, obesity, family history of PE, African American race, low socioeconomic status, age over 35 years, previous adverse pregnancy outcome, and more than a ten-year pregnancy interval). Similar recommendations with slight differences were also proposed by the National Institute for Health and Care Excellence (NICE) [2] Available online: www.nice.org.uk/guidance/ng133 (accessed on 4 October 2021).

Recently, Tong et al. [3] demonstrated that an aspirin dose greater than 100 mg had still been beneficial to prevent PE [4]. Similarly, Meher et al. [5] reported that even starting the aspirin treatment after 16 weeks of gestation might have been useful to prevent PE [4]. However, most of other preventive measures such as magnesium, fish oil, vitamins C, D, and E supplementation fails to bring a benefit in prevention of PE [4,6].

It is evident that only a combination of markers into multiparametric models may provide a useful predictive tool for PE screening applicable in the routine praxis [7].

Currently, an algorithm for the prediction of PE in the first trimester of gestation (between 11 and 14 weeks) developed by the Fetal Medicine Foundation (FMF) has usually been used by most Fetal Medicine Centers over the world. This model for prediction of PE is based on the combination of maternal history, biophysical markers (body mass index, and mean arterial blood pressure), biochemical markers (pregnancy-associated plasma protein-A, and placental growth factor), and Doppler ultrasound parameter (mean uterine artery pulsatility index) [8,9,10,11]. Most Fetal Medicine Centres have been using the Astraia Obstetrics program that was developed in close collaboration with the FMF on the basis of the FMF risk algorithm [12]. It calculates risks for trisomy 21, 18, and 13, fetal growth restriction, and preterm delivery in the first trimester, as well as the risk for PE in all three trimesters. Women identified to be at a high risk for preterm PE according to the FMF algorithm are recommended to receive daily low-dose aspirin (ASA) from 11 to 14 weeks of gestation to 37 weeks of gestation, which may significantly reduce the incidence of preterm PE [13,14].

Chen et al. [15] introduced another risk model to predict hypertensive disorders of pregnancy (GH or PE) including classical markers used during early gestation to identify fetal aneuploidies such as pregnancy-associated plasma protein-A, free beta-human chorionic gonadotropin, and fetal nuchal translucency. However, this model has not yet been robustly tested.

The aims of the current study were to explore if the profiling of selected cardiovascular disease-associated microRNAs [16,17] in the whole peripheral venous blood during the early stages of gestation would be able to detect the presence of chronic hypertension, a high risk factor for the development of PE, and to predict subsequent development of hypertensive pregnancy-related complications such as gestational hypertension (GH) or PE in normotensive women.

We were also interested if the first trimester profiling of cardiovascular disease-associated microRNAs in the whole peripheral venous blood would have any impact on the improvement of the detection rate of PE calculated by the usage of the Astraia Obstetrics program and the FMF risk algorithm [12].

To date, no data on the first trimester microRNA profiling in the whole peripheral venous blood (whole blood cell lysates) is available in women with chronic hypertension and normotensive women with subsequent development of pregnancy-related complication such as GH. Little information on the first trimester microRNA profiling in maternal peripheral blood buffy coat or mononuclear cell samples in women with subsequent development of PE is also available [18,19,20].

## 2. Materials and Methods

### 2.1. Patients Cohort

The retrospective case-control study involving singleton pregnancies of Caucasian descent only was performed. The whole peripheral venous blood samples (EDTA) were collected from 6440 pregnant women undergoing the first trimester screening at 10–13 weeks of gestation in the Institute for the Care of Mother and Child, Prague, Czech Republic, within the period November 2012 to March 2020. Only the samples from women (*n* = 4469) who had been followed-up and delivered in the Institute for the Care of Mother and Child, Prague, Czech Republic, were used for the study. Finally, out of 4469 women with complete medical records, 29 women were confirmed to have diagnosis of chronic hypertension, 83 non-hypertensive women later developed GH, and in 66 non-hypertensive women PE with or without FGR was later diagnosed. In detail, 17 women developed mild PE and 49 women were diagnosed with severe PE. A total of 14 preeclamptic women had early onset of the disease, before 34 weeks of gestation (early PE), and 52 women had late onset of the disease, after 34 weeks of gestation (late PE). 

The control group (*n* = 80), women with normal course of gestation delivering healthy infants after 37 weeks of gestation weighting > 2500 g, was selected on the basis of equal gestational age and equal age of women at the time of sampling and equal biological sample storage times. 

The clinical characteristics of pregnant women are demonstrated in Table 1.

Informed consent was signed by all women participating in the study. The Ethics Committee of the Institute for the Care of the Mother and Child and The Ethics Committee of the Third Faculty of Medicine, Charles University approved the study (Implication of microRNAs in maternal circulation for diagnosis and prediction of placental insufficiency, date of approval: 7 April 2011). All procedures were in compliance with the Helsinki Declaration of 1975, as revised in 2000.

### 2.2. First Trimester Risk Analysis

Astraia Obstetrics Program (Astraia software GmbH, Germany) was developed in close collaboration with the Fetal Medicine Foundation. The FMF risk algorithm [12] calculates risks for trisomy 21, 18, and 13, fetal growth restriction, and preterm delivery in the first trimester of gestation, as well as the risk for preeclampsia in all three trimesters.

### 2.3. Processing of Samples

In brief, small RNAs enriched total RNA was isolated from maternal whole peripheral venous blood cell lysates using a mirVana microRNA Isolation kit (Ambion, Austin, TX, USA). Cardiovascular disease-associated microRNAs (miR-1-3p, miR-16-5p, miR-17-5p, miR-20a-5p, miR-20b-5p, miR-21-5p, miR-23a-3p, miR-24-3p, miR-26a-5p, miR-29a-3p, miR-92a-3p, miR-100-5p, miR-103a-3p, miR-125b-5p, miR-126-3p, miR-130b-3p, miR-133a-3p, miR-143-3p, miR-145-5p, miR-146-5p, miR-155-5p, miR-181a-5p, miR-195-5p, miR-199a-5p, miR-210-3p, miR-221-3p, miR-342-3p, miR-499a-5p, and miR-574-3p) were reverse transcribed into cDNA by TaqMan MicroRNA Assays containing miRNA-specific stem loop primers and the TaqMan MicroRNA Reverse Transcription Kit (Applied Biosystems, Branchburg, NJ, USA) in a total reaction volume of 10 µL [16,17]. A total of 3 µL of cDNA were mixed with the components of TaqMan MicroRNA Assays containing specific primers and the TaqMan MGB probes and the components of the TaqMan Universal PCR Master Mix (Applied Biosystems, Branchburg, NJ, USA) in a total reaction volume of 15 µL. Reverse transcription and real-time qPCR were performed on 7500 Real-Time PCR System using TaqMan PCR conditions set-up in the TaqMan guidelines [16,17]. 

The microRNA expression was assessed using the comparative Ct method [21]. The normalization factor [22] (geometric mean of Ct values of selected endogenous controls: RNU58A and RNU38B) was used to normalize microRNA expression data [16,17]. 

### 2.4. Statistical Analysis

With regard to non-normal distribution of the data microRNA gene expression was compared between pregnant women with chronic hypertension, normotensive women who subsequently developed hypertensive pregnancy-related disorders (GH or PE), and normotensive women with normal course of gestation using the Kruskal–Wallis one-way analysis of variance. Subsequently, post-hoc test for the comparison among multiple groups and Benjamini–Hochberg correction for multiple comparisons were applied [23] (Table 2). The Benjamini–Hochberg method controls the False Discovery Rate (FDR) using sequential modified Bonferroni correction for multiple hypothesis testing. While the Bonferroni correction relies on the Family Wise Error Rate (FWER), Benjamini and Hochberg introduced the idea of a FDR to control for multiple hypotheses testing. In the statistical context, discovery refers to the rejection of a hypothesis. Therefore, a false discovery is an incorrect rejection of a hypothesis and the FDR is the likelihood such a rejection occurs. Controlling the FDR instead of the FWER is less stringent and increases the method’s power. As a result, more hypotheses may be rejected and more discoveries may be made. In the Benjamini–Hochberg method, hypotheses are first ordered and then rejected or accepted based on their p-values. A p-value is a data point for each hypothesis describing the likelihood of an observation based on a probability distribution [24].

Box plots produced using the Statistica software (version 9.0; StatSoft, Inc., Tulsa, OK, USA) display the median, the 75th and 25th percentiles (the upper and lower limits of the boxes), the maximum and minimum values that are no more than 1.5 times the span of the interquartile range (the upper and lower whiskers), outliers (circles), and extremes (asterisks), respectively. 

Receivers operating characteristic (ROC) curve analyses state the areas under the curves (AUC), the best cut-off points related sensitivities, specificities, positive and negative likelihood ratios (LR+, LR−), sensitivities at 10.0% false positive rate (FPR), respectively (MedCalc Software bvba, Ostend, Belgium). To select the optimal microRNA combinations logistic regression with subsequent ROC curve analyses were applied (MedCalc Software bvba, Ostend, Belgium).

Correlation between variables was calculated using the Spearman’s rank correlation coefficient (ρ). If the correlation coefficient values ranged within <0; 0.5>, there was a weak positive correlation. If it was within the interval <−0.5; 0>, there was a weak negative correlation.

## 3. Results

Gene expression of cardiovascular disease-associated microRNAs in peripheral blood leukocytes was compared in early stages of gestation (within 10 to 13 weeks) between women with chronic hypertension, normotensive women who subsequently developed hypertensive pregnancy-related disorders (GH or PE wo/w FGR) and normotensive women with normal course of gestation. Appendix A displays only statistical significant data after Benjamini–Hochberg correction for multiple comparisons applied after the Kruskal–Wallis test. Appendix A displays statistical non-significant data after Benjamini–Hochberg correction. Just the results that reached a statistical significance after Benjamini–Hochberg correction for multiple comparisons applied after the Kruskal-Wallis test are discussed below (Table 2). To interpret the experimental data new cut-off point *p*-values were set-up. Significant results after the Benjamini-Hochberg correction are marked by asterisks for appropriate significance levels (* for α = 0.05, ** for α = 0.01, and *** for α = 0.001).

### 3.1. Cardiovascular Disease-Associated MicroRNAs Are Dysregulated in Early Stages of Gestation in Women with Chronic Hypertension and Normotensive Women Subsequently Developing GH or PE wo/w FGR

The gene expression of miR-1-3p (*p* < 0.001 ***) and miR-195-5p (*p* < 0.001 ***) was uniquely upregulated in pregnant women with chronic hypertension. Increased levels of miR-20a-5p (*p* < 0.001 ***, *p* = 0.001 **), miR-146a-5p (*p* < 0.001 ***, *p* < 0.001 ***), and miR-155-5p (*p* < 0.001 ***, *p* = 0.001 **) were detected during the first trimester of gestation in pregnant women with chronic hypertension and in pregnant women that subsequently developed PE wo/w FGR (Appendix A). Upregulation of miR-181a-5p was a common phenomenon of women with chronic hypertension (*p* = 0.004 *), and those ones subsequently developing GH (*p* = 0.007 *) or PE wo/w FGR (*p* < 0.001 ***). MiR-143-3p (*p* = 0.002 **), miR-145-5p (*p* < 0.001 ***), and miR-574-3p (*p* = 0.007 *) represented unique upregulated biomarkers in women with subsequent onset of PE wo/w FGR (Appendix A).

The ROC curve analyses showed very good sensitivities at 10.0% FPR for miR-1-3p (51.72%), miR-20a-5p (44.83%), and miR-146a-5p (65.52%) in pregnant women with chronic hypertension. A proportion of pregnant women who subsequently developed PE wo/w FGR had upregulated expression profile of miR-20a-5p (33.33%), miR-143-3p (24.24%), miR-145-5p (21.21%), miR-146a-5p (42.42%), miR-181a-5p (40.91%), and miR-574-3p (27.27%) at 10.0% FPR during the first trimester of gestation (Appendix A). 

Upregulation of miR-181a-5p was also present in early stages of gestation in 22.89% of women at 10.0% FPR, who subsequently developed GH (Appendix A).

Other microRNAs (miR-155-5p, miR-195-5p) showed the poor sensitivity at 10.0% FPR; therefore, these particular microRNAs were not further considered as early gestation biomarkers distinguishing between normal and pathological course of gestation (Appendix A).

### 3.2. The High Accuracy of First Trimester Combined MicroRNA Screening to Differentiate between Women with Chronic Hypertension and Normotensive Women with Normal Course of Gestation

The combined screening of miR-1-3p, miR-20a-5p, and miR-146a-5p (AUC 0.865, *p* < 0.001, 72.41% sensitivity, 90.0% specificity, cut off > 0.3236) revealed that at 10.0% FPR 72.41% women with chronic hypertension had aberrant microRNA expression profile in early stages of gestation (Figure 1). According to NICE 2019 guidelines [2] and ACOG 2018 guidelines [1] women with chronic hypertension in anamnesis are considered as high risky patients for subsequent development of PE and should be given ASA as early as possible to decrease the risk of later development of the disease (75–150 mg ASA from 12th week of gestation according to NICE 2019 recommendation, and 81 mg ASA before 16th week of gestation according to ACOG 2018 recommendation). In our group of women ASA was given to 15 out of 29 women (51.72%) with chronic hypertension. ASA was administered if not contraindicated mainly to those patients who had been pregnant after a period when the NICE and ACOG recommendations came into awareness of scientific public. Four out of 29 women (13.79%) with chronic hypertension had signs of PE, four women (13.79%) decompensated hypertension during gestation despite intensive antihypertensive treatment, four women (13.79%) delivered at term with grade I hypertension, one woman (3.45%) delivered at term with grade II hypertension, 14 women (48.28%) delivered at term with normal blood pressure, and two women (6.90%) delivered preterm for other reasons.

### 3.3. The Very Good Accuracy of First Trimester Combined MicroRNA Screening to Differentiate between Normotensive Women with Subsequent Development of PE wo/w FGR and Normal Course of Gestation

The combined screening of miR-20a-5p, miR-143-3p, miR-145-5p, miR-146a-5p, miR-181a-5p, and miR-574-3p was superior over using individual microRNA biomarkers or different microRNA combinations, since it was able to detect at 10.0% FPR in early stages of gestation aberrant microRNA expression profile in 48.48% normotensive women subsequently developing PE wo/w FGR (AUC 0.730, *p* < 0.001, 66.67% sensitivity, 75.0% specificity, cut off > 0.421) (Figure 2).

In our group of women the first trimester screening for PE and/or FGR using the criteria of the Fetal Medicine Foundation [12] detected 22 out of 66 normotensive women subsequently developing PE wo/w FGR (33.33%). ASA was subsequently given to 20 women based on the positive results of the first trimester screening for PE and/or FGR. In addition, two women with negative results of the first trimester screening for PE and/or FGR were given ASA as well for other reasons (mainly with regard to the history of PE in anamnesis).

The comparison of the predictive results of the routine first trimester screening for PE and/or FGR based on the criteria of the Fetal Medicine Foundation [12] and the first trimester screening for PE wo/w FGR using a panel of six cardiovascular disease-associated microRNAs only revealed that the detection rate of PE increased 1.45-fold (48.48% vs. 33.33%).

### 3.4. Correlation between First Trimester MicroRNA Screening and the Routine First Trimester Predictive Markers for PE and/or FGR

In the total group of women (women with both normal and complicated pregnancies), a weak positive correlation between microRNA gene expression (miR-20a-5p, miR-103a-3p, miR-125b-5p, miR-143-3p, miR-145-5p, miR-146a-5p, miR-155-5p, miR-181a-5p, and miR-574-3p) in the whole maternal peripheral blood and the mean arterial pressure (mmHg) in early stages of gestation was detected. Nevertheless, in case of MAP (MoM) only a weak positive correlation with miR-155-5p levels in the whole maternal peripheral blood was observed (Appendix A).

In parallel, a weak negative correlation between miR-92a-3p levels in the whole maternal peripheral blood and serum PAPP-A levels (IU/L) was found during the first trimester of gestation. However, more microRNAs (miR-16-5p, miR-146a-5p, miR-155-5p, miR-210-3p, and miR-221-3p) showed a weak negative correlation with serum PAPP-A levels (MoM) (Appendix A).

No correlation between microRNA gene expression in the whole maternal peripheral blood and the uterine artery pulsatility index (real values and/or MoM) and serum PIGF levels (pg/mL and/or MoM) was observed in early stages of gestation.

## 4. Discussion

Gene expression of 29 selected cardiovascular disease-associated microRNAs was compared in peripheral blood leukocytes during the first trimester of gestation (within 10 to 13 weeks) between women with chronic hypertension, normotensive women who subsequently developed hypertensive pregnancy-related disorders (GH or PE wo/w FGR) and normotensive women with normal course of gestation.

Up-regulation of miR-1-3p and miR-195-5p was observed in early stages of gestation just in pregnant women with chronic hypertension. Up-regulation of miR-20a-5p, miR-146a-5p, and miR-155-5p was a common feature of pregnant women with chronic hypertension and those ones that subsequently developed PE wo/w FGR. MiR-181a-5p was up-regulated in pregnant women with chronic hypertension, and pregnancies that subsequently developed GH or PE wo/w FGR. Up-regulation of miR-143-3p, miR-145-5p, and miR-574-3p represented a unique early feature of subsequent development of PE wo/w FGR.

To our knowledge, limited data on prediction of adverse pregnancy outcome using microRNA profiling in maternal leukocytes during the first trimester of gestation was reported. Initially, Winger et al. [18] reported prediction of miscarriage and the development of late PE in a pilot study using eight miscarriage and 12 PE maternal peripheral blood mononuclear cell samples and a panel of 30 microRNAs. Later, the group of Winger et al. [19] introduced a reduced panel of seven selected microRNAs (miR-1, miR-133b, miR-199a-5p, miR-1267, miR-1229, miR-223, and miR-148a-3p) to predict successfully miscarriage and PE using again maternal peripheral blood mononuclear cells in a period < six weeks post-implantation in a small scale study involving eight miscarriage and 12 PE patients. Finally, the group of Winger et al. [20] demonstrated the ability to predict with a high accuracy the development of PE using an original panel of 30 microRNAs (miR-1267, miR-148, miR-196a, miR-33a, miR-575, miR-582, miR-210, miR-16, miR-1229, miR-223, miR-133b, miR-155, miR-146a, miR-181a, miR-301a, miR-340, miR-30e-3p, miR-132, miR-1244, miR-671, miR-7-5p, miR-1, miR-144-3p, miR-193a-3p, miR-199a, miR-199b-5p, miR-219, miR-221, miR-424, and miR-513) in buffy coat blood samples between 11 and 13 weeks of gestation in a cohort involving eight PE patients and 40 pregnancies with normal course of gestation. From the panel of 30 microRNAs used by Winger et al. [18,20] we had eight mutual microRNA biomarkers. After detailed exploration of miRBase ID corresponding to TaqMan Assays miRNAs used by Winger et al. [18,20] we found out that in most cases we used the same microRNA biomarkers (miR-210-3p, miR-16-5p, miR-146a-5p, miR-181a-5p, miR-1-3p, miR-199a-5p, miR-221-3p). Nevertheless, in one case they used stem-loop microRNA sequence (miR-155), whereas we used mature microRNA sequence (miR-155-5p).

Concerning miR-210-3p, miR-16-5p, miR-155-5p, miR-146a-5p, and miR-181a-5p our expression data are inconsistent with the data of Winger et al. [18,19]. While we observed no difference between women subsequently developing PE wo/w FGR and women with normal course of gestation, they observed significantly increased levels of miR-210-3p and miR-16-5p in women with normal course of gestation. On the other hand, while we observed significantly increased levels of miR-155-5p, miR-146a-5p, and miR-181a-5p in women subsequently developing PE wo/w FGR, they found out no difference in microRNA expression between women subsequently developing PE and women with normal course of gestation [20]. Concerning miR-1-3p, miR-199a-5p, and miR-221-3p the data cannot be mutually compared, since microRNA expression data are not available in a study of Winger et al. [20]. These discrepancies may be caused by low number of PE patients involved in the studies of Winger et al. [18,19,20], while our study involved 66 PE wo/w FGR and 80 normal gestation patients, which already represents a moderate scale study providing more accurate results.

Other studies focused more likely on the search of circulating microRNA biomarkers detectable in maternal plasma or serum with predictive value for PE and demonstrated a whole range of diverse microRNAs with altered expression during the first trimester of gestation in women subsequently developing PE involving miR-1233 (up-regulated), miR-520a (up-regulated), miR-210 (up-regulated), miR-144 (down-regulated) [25], miR-520g (up-regulated) [26], miR-942 (down-regulated) [27], miR-517-5p (up-regulated) [28], miR-423-5p (up-regulated) [29], miR-182 (down-regulated), miR-10b (down-regulated), miR-25 (down-regulated), miR-4433b (up-regulated), miR-99b (down-regulated), miR-143 (down-regulated), miR-151a (down-regulated), miR-191 (down-regulated), miR-146b (down-regulated), miR-221 (up-regulated), let-7g (up-regulated) [30], miR-23b-5p (down-regulated), miR-99b-5p (down-regulated) [31], and miR-125b (up-regulated) [32]. Nevertheless, it does not make sense to compare intracellular and extracellular microRNA expression profiles, since they can differ for various reasons. Furthermore, previous study used stem-loop microRNA sequences (miR-143 and miR-221) [30], whereas we used mature microRNA sequences (miR-143-3p and miR-221-3p) to predict PE wo/w FGR. Maternal plasma exosomal profiling of selected C19MC microRNAs also revealed novel down-regulated microRNA biomarkers during the first trimester of gestation (miR-517-5p, miR-520a-5p, and miR-525-5p) for women destinated to develop GH or PE [33].

Concerning pregnant women with chronic hypertension, our data are consistent with studies of other researchers who also observed increased expression of miR-1 in peripheral blood mononuclear cells in patients with essential hypertension with and without clinical indices of left ventricular hypertrophy [34,35].Similarly, our data correspond with data of another research group that observed significantly increased levels of miR-20a-5p in peripheral blood mononuclear cells of hypertensive females when compared with normotensive females, however the studied group of females was of African American descent only [36]. Recently, we have observed increased postpartal expression or a trend towards increased postpartal expression of miR-17-5p, miR-24-3p, miR-92a-3p, miR-126-3p, miR-130b-3p, miR-181a-5p, and miR-210-3p in peripheral blood mononuclear cells in mothers with systolic hypertension when the comparison to mothers with normal systolic blood pressure values regardless of a history of gestation was performed [16]. However, from the current study implies that at least up-regulation of miR-126-3p (*p* = 0.016), miR-130b-3p (*p* = 0.086), and miR-181a-5p (*p* = 0.003 *) may already be present in early stages of gestation in women with chronic hypertension; however, it does not reach statistical significance in some microRNAs, when Benjamini–Hochberg correction for multiple comparisons was used. In addition, miR-181a-5p was demonstrated to suppress mRNA encoded by hypertension-related *ADM* and *SH2B3* genes [37]. Concerning miR-146a-5p [38,39], miR-155-5p [40], and miR-195-5p [41], only the studies on expression microRNA levels in plasma or serum of hypertensive patients are available. However, once again, it does not make sense to compare intracellular and circulating microRNA expression profiles, since they can differ for various reasons.

Subsequently, we analyzed microRNA expression profiles in the whole maternal peripheral blood in relation to the routine first trimester predictive markers for PE and/or FGR (MAP, UtA-PI, serum PAPP-A, and PIGF levels). Correlations between some microRNAs and MAP (miR-20a-5p, miR-103a-3p, miR-125b-5p, miR-143-3p, miR-145-5p, miR-146a-5p, miR-155-5p, miR-181a-5p, and miR-574-3p) and serum PAPP-A levels (miR-16-5p, miR-92a-3p, miR-146a-5p, miR-155-5p, miR-210-3p, and miR-221-3p) were detected. Nevertheless, these were only weak positive and/or weak negative correlations.

As increased levels of MAP [8,9,10,12], so decreased serum levels of PAPP-A [11,15,42,43] during the first trimester of gestation were observed to be significantly associated with subsequent onset of PE. Newly, the upregulation of some microRNAs in the whole maternal peripheral blood (miR-20a-5p, miR-143-3p, miR-145-5p, miR-146a-5p, miR-155-5p, miR-181a-5p, and miR-574-3p) during the first trimester of gestation was observed to be significantly associated with the presence of chronic hypertension or subsequent onset of GH or PE.

## 5. Conclusions

A significant proportion of patients with chronic hypertension had up-regulated miR-1-3p expression profile. Interestingly, similar miR-20a-5p and miR-146a-5p expression profiles were identified in a significant proportion of pregnant women with chronic hypertension and normotensive women subsequently developing PE wo/w FGR. Parallel, up-regulation of miR-181a-5p was detected in a significant proportion of normotensive women subsequently developing PE wo/w FGR and in a smaller proportion of normotensive women subsequently developing GH. On the other hand, a lesser proportion of women with subsequent onset of PE wo/w FGR have also upregulated microRNA expression profile (miR-143-3p, miR-145-5p, and miR-574-3p).

The combination of upregulated microRNA biomarkers (miR-20a-5p, miR-143-3p, miR-145-5p, miR-146a-5p, miR-181a-5p, and miR-574-3p) can predict the later occurrence of PE wo/w FGR in nearly one-half of cases in early stages of gestation. In addition, the combination of upregulated microRNA biomarkers (miR-1-3p, miR-20a-5p, and miR-146a-5p) is able to identify most pregnancies with chronic hypertension in early stages of gestation.

Consecutive large-scale studies are needed to verify our pilot data. However, the implementation of that kind of large-scale studies is highly demanding, since thousands of samples have to be collected during the early stages of gestation to acquire adequate number of samples from women with the subsequent onset of GH or PE wo/w FGR during the third trimester of gestation. For the purpose of this study we have consecutively collected the samples for nearly eight years to acquire sufficient number of samples from women that later developed relevant pregnancy-related complications (83 GH and 66 PE wo/w FGR). Nevertheless, it seems that microRNAs represent promising biomarkers with very good diagnostical potential to be implemented into current first trimester screening program to predict later occurrence of PE wo/w FGR. The comparison of the predictive results of the routine first trimester screening for PE and/or FGR based on the criteria of the Fetal Medicine Foundation [12] and the first trimester screening for PE wo/w FGR using a panel of six cardiovascular disease-associated microRNAs only revealed that the detection rate of PE was increased 1.45-fold.

## 6. Patents

National patent application—Industrial Property Office, Czech Republic (Patent n. PV 2021-562).

## Figures and Tables

**Figure 1 biomedicines-10-00256-f001:**
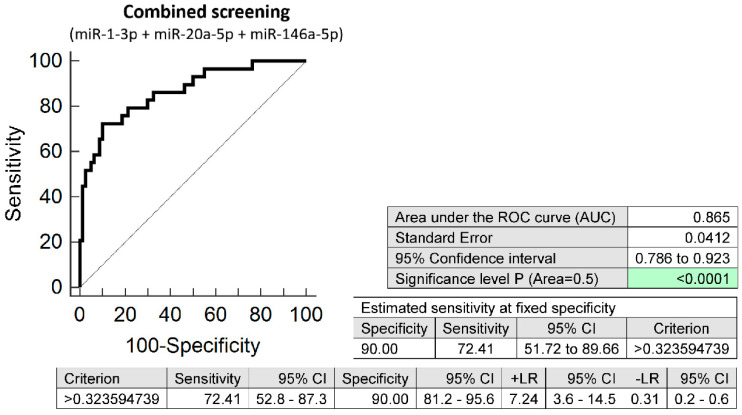
ROC analysis—the combination of microRNA biomarkers. The combination of miR-1-3p, miR-20a-5p, and miR-146a-5p showed that at 10.0% FPR 72.41% women with chronic hypertension had aberrant microRNA expression profile in peripheral blood leukocytes in early stages of gestation. Green colour highlighted significant *p*-value.

**Figure 2 biomedicines-10-00256-f002:**
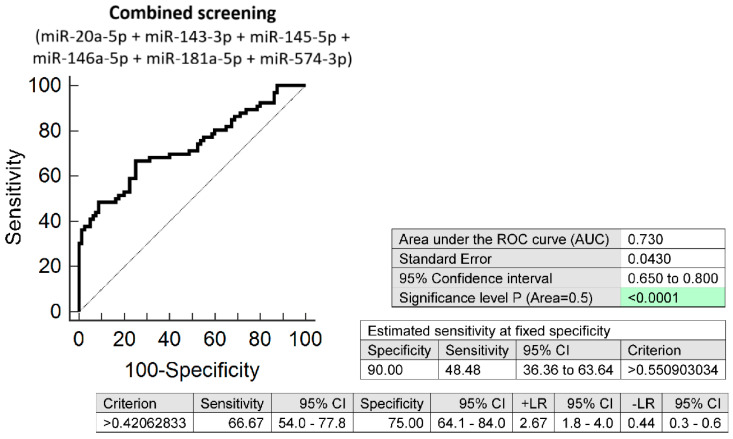
ROC analysis—the combination of microRNA biomarkers. The combination of miR-20a-5p, miR-143-3p, miR-145-5p, miR-146a-5p, miR-181a-5p, and miR-574-3p showed that at 10.0% FPR 48.48% normotensive women subsequently developing PE w/wo FGR had aberrant microRNA expression profile in peripheral blood leukocytes in early stages of gestation. Green colour highlighted significant *p*-value.

**Table 1 biomedicines-10-00256-t001:** Clinical characteristics of the controls and complicated pregnancies.

	Normal Pregnancies (*n* = 80)	CHH (*n* = 29)	GH (*n* = 83)	PE wo/w FGR (*n* = 66)	*p*-Value ^1^	*p*-Value ^2^	*p*-Value ^3^
*Maternal characteristics*							
Maternal age (years)	32.0 (29.0–34.25)	34.0 (32.0–39.0)	34.0 (30.0–36.5)	31.0 (28.25–36.0)	0.032	0.445	1.000
Advanced maternal age (≥35 years old)	20 (25%)	14 (48.28%)	34 (40.96%)	22 (33.33%)	0.020	0.030	0.268
Caucasian ethnic group	80 (100%)	29 (100%)	83 (100%)	66 (100%)	-	-	-
Prepregnancy BMI (kg/m^2^)	21.28 (20.54–23.55)	25.56 (21.67–31.31)	24.8 (22.65–30.76)	23.98 (21.37–28.17)	<0.001	<0.001	<0.001
Diabetes mellitus (T1DM, T2DM)	0 (0%)	2 (6.90%)	6 (7.23%)	4 (6.06%)	-	-	-
Autoimmune diseases (SLE/APS/RA)	0 (0%)	1 (3.45%)	3 (3.61%)	2 (3.03%)	-	-	-
Parity							
Nulliparous	40 (50.0%)	15 (51.72%)	55 (66.26%)	52 (78.79%)	1.000	0.035	<0.001
Parous with no previous PE	40 (50.0%)	11 (37.93%)	26 (31.33%)	8 (12.12%)	-	-	-
Parous with previous PE	0 (0%)	3 (10.34%)	2 (2.41%)	6 (9.09%)	-	-	-
ART (IVF/ICSI/other)	2 (2.5%)	7 (24.14%)	15 (18.07%)	17 (25.76%)	<0.001	0.001	<0.001
Smoking during pregnancy	2 (2.5%)	2 (6.90%)	3 (3.61%)	3 (4.55%)	0.281	0.680	0.499
*Pregnancy details* (*First trimester of gestation*)							
Gestational age at sampling (weeks)	10.29 (10.14–10.57)	10.57 (10.14–10.71)	10.29 (10.0–10.71)	10.43 (10.03–11.0)	0.468	1.000	1.000
MAP (mmHg)	88.75 (84.54–95.04)	100.83 (96.5–112.75)	99.67 (95.62–104.83)	96.0 (91.0–100.08)	<0.001	<0.001	<0.001
MAP (MoM)	1.05 (1.01–1.10)	1.16 (1.11–1.24)	1.14 (1.11–1.19)	1.13 (1.07–1.17)	<0.001	<0.001	0.017
Mean UtA-PI	1.39 (1.11–1.68)	1.38 (1.15–1.78)	1.54 (1.24–1.95)	1.55 (1.13–1.94)	0.195	1.000	1.000
Mean UtA-PI (MoM)	0.90 (0.73–1.10)	0.92 (0.70–1.16)	1.02 (0.82–1.27)	1.0 (0.71–1.26)	0.111	1.000	1.000
PIGF serum levels (pg/mL)	27.1 (21.6–34.3)	24.6 (19.45–36.37)	25.0 (20.8–29.6)	22.8 (17.8–29.2)	1.000	0.442	0.040
PIGF serum levels (MoM)	1.04 (0.85–1.31)	0.94 (0.82–1.19)	1.06 (0.87–1.23)	0.96 (0.7–1.20)	1.000	1.000	0.294
PAPP-A serum levels (IU/L)	1.49 (1.09–2.36)	1.19 (0.60–2.66)	1.10 (0.65–1.95)	1.28 (0.73–2.05)	0.575	0.016	0.214
PAPP-A serum levels (MoM)	1.17 (0.82–1.54)	0.92 (0.66–1.35)	1.07 (0.70–1.54)	0.93 (0.61–1.38)	0.917	1.000	0.118
Screen-positive for PE and/or FGR by FMF algorithm	0 (0%)	16 (55.17%)	29 (34.94%)	22 (33.33%)	-	-	-
Aspirin intake during pregnancy	0 (0%)	15 (51.72%)	26 (31.33%)	22 (33.33%)	-	-	-
*Pregnancy details* (*At delivery*)							
Systolic blood pressure (mmHg)	122 (115–131)	136.5 (126.75–146.25)	147.5 (137.75–154.75)	155 (145–165.75)	<0.001	<0.001	<0.001
Diastolic blood pressure (mmHg)	76.5 (70.75–81.25)	85.0 (80–95.25)	94.5 (89–99)	100 (92.25–103)	<0.001	<0.001	<0.001
Gestational age at delivery (weeks)	40.07 (39.10–40.60)	38.78 (37.82–40.03)	39.14 (38.36–40.07)	37.07 (34.75–38.25)	0.002	0.005	<0.001
Delivery at gestational age < 37 weeks	0 (0%)	4 (13.79%)	10 (12.05%)	33 (50.0%)	-	-	-
BMI (kg/m^2^)	26.66 (25.11–28.81)	29.81 (26.27–33.10)	30.67 (28.07–35.66)	29.83 (26.64–33.96)	0.007	<0.001	<0.001
Weight gain during pregnancy (kg)	14 (12–17.75)	10 (7.75–12.725)	14 (10–18)	14 (10–19)	0.001	1.000	1.000
Fetal birth weight (grams)	3470 (3290–3690)	3240 (2965–3475)	3370 (3110–3785)	2645 (2065–3222.5)	0.996	0.076	<0.001
Fetal sex							
Boy	40 (50.0%)	16 (55.17%)	47 (56.63%)	29 (43.94%)	0.633	0.396	0.465
Girl	40 (50.0%)	13 (44.83%)	36 (43.37%)	37 (56.06%)			
Mode of delivery							
Vaginal	69 (86.25%)	14 (48.28%)	48 (57.83%)	18 (27.27%)	<0.001	<0.001	<0.001
CS	11 (13.75%)	15 (51.72%)	35 (42.17%)	48 (72.73%)			

Continuous variables, compared using the Kruskal–Wallis test, are presented as median (IQR; interquartile range). Categorical variables, presented as number (percent), were compared using Chi-squared test. *p*-value ^1,2,3^: the comparison among normal pregnancies and chronic hypertension, gestational hypertension, or preeclampsia, respectively. PE, preeclampsia; FGR, fetal growth restriction; CHH, chronic hypertension; GH, gestational hypertension; BMI, body mass index; T1DM, type 1 diabetes mellitus; T2DM, type 2 diabetes mellitus; SLE, systemic lupus erythematosus; APS, antiphospholipid syndrome; RA, rheumatoid arthritis; ART, assisted reproductive technology; IVF, in vitro fertilization; ICSI, intracytoplasmic sperm injection; MAP, mean arterial pressure; UtA-PI, uterine artery pulsatility index; PIGF, placental growth factor; PAPP-A, pregnancy-associated plasma protein-A; FMF, Fetal Medicine Foundation; CS, Caesarean section.

**Table 2 biomedicines-10-00256-t002:** Benjamini–Hochberg correction for multiple comparisons: Comparison of microRNA gene expression between various groups of pregnant women (normal pregnancies vs. chronic hypertension vs. GH vs. PE).

K	i	Alpha = 0.05	Alpha = 0.01	Alpha = 0.001
**6**		**0.05**	**0.01**	**0.001**
	**1**	0.008	0.002	0.000
	**2**	0.017	0.003	0.000
	**3**	0.025	0.005	0.001
	**4**	0.033	0.007	0.001
	**5**	0.042	0.008	0.001
	**6**	0.050	0.010	0.001

## Data Availability

The data presented in this study are available on request from the corresponding author. The data are not publicly available due to rights reserved by funding supporters.

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
