# Peer review of "Cardiovascular Disease-Associated MicroRNA Dysregulation during the First Trimester of Gestation in Women with Chronic Hypertension and Normotensive Women Subsequently Developing Gestational Hypertension or Preeclampsia with or without Fetal Growth Restriction"

_biomedicines, 2022, doi:10.3390/biomedicines10020256_

Round 1
Reviewer 1 Report
The article is very interesting, once hypertensive pregnancy-related complications, manly gestational hypertension or preeclampsia. remain without an efficient therapy for their treatment, so an early detection may represent a great advance for science. Moreover, microRNAs may represent a promising biomarkers with very good diagnostical potential to be implemented into current first trimester screening programme to predict later occurrence of PE wo/w FGR.
General comments:
- The abstracted is good.
- The introduction could be improved, with a better explanation of the ineffective therapy that still exists, please see the review (Pre-Eclampsia and Eclampsia: An Update on the Pharmacological Treatment Applied in Portugal, DOI: 10.3390/jcdd5010003). It could also explain better the pathologies (A risk model of prenatal screening markers in first trimester for predicting hypertensive disorders of pregnancy, doi: 10.1007/s13167-020-00212-3; Prediction of pre-eclampsia, DOI: 10.1177/1753495X20984015)
- In the 2.3. Processing of samples, I did not understand where the blood comes from? from the mothers? this is a retrospective study, so this collection must be very well defined. Please the collection procedure
- Please explain the use Benjamini-Hochberg correction.
- In the discussion the author could discuss more the clinical parameters, although the authors put them later they do not make the relation of them with the microRNAs. Please include this part.
Reviewer 2 Report
This is a good organized paper.
The article takes up an important topic and issues. The work is very interesting. Discussion and conclusions correct. Good organization of the work text. This is a good manuscript and well done study. I recommend this manuscript for publication
Author Response
Than you very much for a nice evaluation of our manuscript.